# Effect of Sacubitril-Valsartan on Quality of Life, Functional and Exercise Capacity in Heart Failure with Preserved Ejection Fraction (HFpEF): A Systematic Review of Randomized Clinical Trials

Advait Vasavada *, Akhil Sadhu, Carla Valencia, Hameeda Fatima, Ijeoma Nwankwo, Mahvish Anam, Shrinkhala Maharjan, Zainab Amjad, Abdelrahman Abaza and Safeera Khan

California Institute of Behavioral Neurosciences and Psychology, Fairfield, CA 94534, USA
* Correspondence: advait2163@gmail.com

**Abstract:** Background: Sacubitril/Valsartan use in heart failure has shown promising results in early trials. However, the effects on the overall functional capacity, exercise capacity, and quality of life are unknown. Aims: We aimed to understand the results of studies that attempted to measure these outcomes that affect the mobility and day-to-day life of these patients. Methods: MEDLINE, PubMed, PubMed Central (PMC), Google Scholar, ClinicalTrials.gov, and ISRCTN were explored to look for clinical trials relevant to the literature. Results: A total of three high-quality randomized controlled trials were discovered that evaluated the effect of sacubitril-valsartan on functional capacity, exercise capacity, or quality of life. All of them were industry-funded and revealed no statistical difference in the mentioned outcomes. No study measured peak oxygen uptake or ventilation/carbon dioxide ratio slope. Conclusion: Sacubitril-valsartan had minimal to no impact on functional capacity, exercise capacity, or quality of life. However, future prospective studies with more sensitive outcome measures should be conducted to validate the findings.

**Keywords:** heart failure with preserved ejection fraction; exercise capacity; functional capacity; quality of life

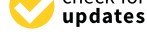



## 1. Introduction

Heart failure with preserved ejection fraction (HFpEF) accounts for almost half of all cases of heart failure [1]. It has been proven that the condition has significant morbidity and mortality, so ignoring the condition just because the ejection fraction is normal is no longer viable [2]. Due to the lack of preclinical models, the development of therapeutics has been a challenge [3]. The condition is primarily related to diastolic dysfunction which is present at rest or induced by stress. The ejection fraction is normal at rest and does not respond to stress, which should appropriately increase [4]. It is also postulated that such patients have increased sensitivity to changes in preload and afterload and this manifests as rapid onset pulmonary edema [5]. The pathophysiology of this condition remains an enigma.

There is some evidence that cardiomyopathy remodeling may be the underlying cause of the condition. Many other mechanisms are defined but most of them are yet to be validated [6].

The criteria for diagnoses also are very heterogeneous among many studies. Hence, this fact makes the condition very difficult to study. Due to the aforementioned reasons, the management of the condition is unclear [7].

The management of heart failure with preserved ejection fraction is emerging with new classes of drugs proving to be very beneficial in treating the condition, particularly sacubitril valsartan. Although the drug sacubitril-valsartan has been most validated in heart failure with reduced ejection fraction, there is little evidence regarding its use in

patients with HFpEF. Recently, there is emerging evidence for the use of sacubitril-valsartan for the treatment of HFpEF and several guidelines also have included the drug in their recommendations, but there is little evidence regarding the effect of the drug on the quality of life, functional capacity, or exercise capacity even though it has shown to reduce hospitalizations [8]. This information is very relevant to the context of the treatment of HFpEF. One of the most common complaints of HFpEF patients is dyspnea and exercise intolerance. Impaired cardiac, pulmonary and peripheral reserve and a few other factors play a role in limiting exercise tolerance in these patients. Since most of the affected population is elderly and the disease burden is increasing at an alarming rate, we need therapies that target the mentioned outcomes [9].

In this systematic review, we explore recent evidence evaluating sacubitril-valsartan or LCZ696 as a part of HFpEF management and discuss pivotal results in the clinical trials, and highlight questions requiring additional inquiry regarding outcomes pertaining to functional capacity, quality of life, and exercise capacity.

## 2. Materials and Methods

We used the PRISMA (Preferred Reporting Items for Systematic Reviews and Meta-analysis) guidelines [10] and principles to design this study.

### 2.1. Search Strategy

We used major research literature databases and search engines such as Google Scholar, MEDLINE, PubMed, PubMed Central (PMC), ClinicalTrials.gov, and ISRCTN to look for clinical trials relevant to the topic of interest. The search was carried out for studies conducted before 21 October 2022. The search was a combination of medical subject headings and keywords. The final search strategy yielding the most results is as follows: ("heart failure with preserved ejection fraction" OR "HfpEF" OR "diastolic heart failure") AND (LCZ696 OR "sacubitril-valsartan" OR Entresto). Other keywords used include: functional capacity, exercise capacity, exercise tolerance, quality of life, clinical trials, randomised controlled trials (RCT), RCT. To find relevant articles, these keywords were combined in varying combinations using Booleans "AND", "OR", and "NOT". Inclusion criteria include papers relevant to the research question, peer-reviewed and non-peer-reviewed clinical trials, papers focusing on the adult population: age > 18 years, studies in grey literature in the English language, papers with full text, and clinical trials. Exclusion criteria included narrative review articles, systematic reviews, short communications, observational studies, and literature that does not measure the quality of life, functional capacity, or exercise capacity. The criteria were strictly adhered to for searching relevant clinical trials.

### 2.2. Analysis of Study Quality/Bias

We critically evaluated all the studies for the risk of bias using standard tools for quality assessment. The Cochrane risk of bias assessment tool was used to evaluate the studies. Two investigators evaluated this independently, and whenever a disagreement occurred, a third investigator's consultation was sought. In none of the studies was there any indication of an imbalance in baseline characteristics. The detailed scoring of all studies for each study is provided in Table 1.

**Table 1.** The Cochrane tool for the quality assessment of clinical trials.

| Study | Random Sequence Generation | Allocation Concealment | Blinding of Participants and Personnel | Incomplete Outcome Data | Selective Reporting | Other Bias |
|---|---|---|---|---|---|---|
| PARAGON Trial (NCT01920711) [11] | + | + | + | + | + | + |
| PARALLAX Trial (NCT03066804) [12] | + | + | + | + | + | + |

**Table 1.** *Cont.*

| Study | Random Sequence Generation | Allocation Concealment | Blinding of Participants and Personnel | Incomplete Outcome Data | Selective Reporting | Other Bias |
|---|---|---|---|---|---|---|
| PARAMOUNT Trail (NCT00887588) [13] | + | + | + | + | + | + |
| Key | | | | | | |
| + | Low risk of bias | | | | | |
| − | High risk of bias | | | | | |
| ? | Unclear bias | | | | | |

### *2.3. Data Extraction*

Two investigators independently extracted data from the eligible studies and examined them for the following: (1) type of study; (2) the number of participants; (3) outcome measures for functional capacity, exercise capacity and quality of life.

### 3. Results

A total of 56,257 articles were identified during the initial search on Google Scholar, MEDLINE, PubMed, PubMed Central (PMC), ClinicalTrials.gov, and ISRCTN. After using relevant filters, keywords, and operators based on our eligibility criteria, 55,770 were discarded.

Two investigators then screened the remaining articles based on titles, abstract, full text, and detailed inclusion-exclusion criteria. Whenever a disagreement occurred, a third investigator was consulted. After the meticulous screening, we were left with 3 clinical trials of interest. The PRISMA flow diagram is depicted in Figure 1 [10,14].

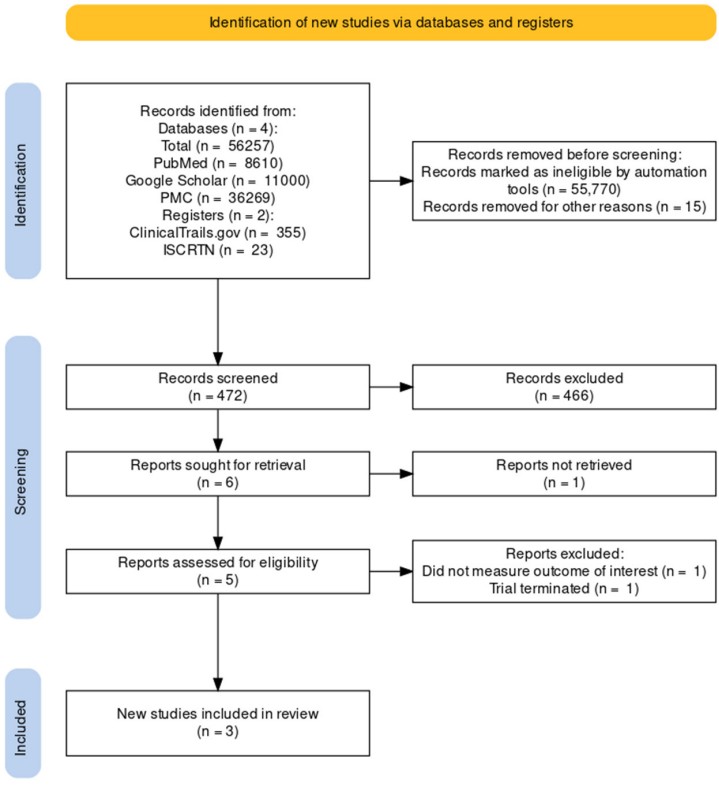

**Figure 1.** PRISMA flowchart.

We reviewed the impact of sacubitril-valsartan on functional capacity, exercise capacity, and quality of life by comparing the outcome measures. All studies were high quality and

were funded by Novartis. Incidentally, we did not find a single study that was performed independently without funding.

A brief description of each study including the trial number, number of patients, type of study, and results regarding the effect of sacubitril-valsartan is listed in Table 2.

**Table 2.** The study characteristics.

| Study | Outcomes | Title | Sponsor | Age | Patients Enrolled | Phases |
|---|---|---|---|---|---|---|
| NCT01920711 [12] | KCCQ, NYHA class | Efficacy and Safety of LCZ696 Compared to Valsartan, on Morbidity and Mortality in Heart Failure Patients With Preserved Ejection Fraction | Novartis Pharmaceuticals | 50 Years and older | 4822 | Phase 3 |
| NCT03066804 [13] | 6MWD, KCCQ, NYHA class, SF36 PCS | A Randomized, Double-blind Controlled Study Comparing LCZ696 to Medical Therapy for Comorbidities in HfpEF Patients | Novartis Pharmaceuticals | 45 Years and older | 2572 | Phase 3 |
| NCT00887588 [14] | KCCQ, NYHA class, Clinical Composite Assessment | LCZ696 Compared to Valsartan in Patients With Chronic Heart Failure and Preserved Left-ventricular Ejection Fraction | Novartis Pharmaceuticals | 40 Years and older | 307 | Phase 2 |

KCCQ: Kansas City Cardiomyopathy Questionnaire, 6MWD: 6 Minute Walk Distance, NYHA: New York Heart Association, SF36-PCS: Short form health survey physical component score.

### 3.1. NCT01920711 [12]

The PARAGON trial mostly focused on the drug's effect on morbidity and mortality in HFpEF. However, it also gave an insight into the change in functional capacity due to the drug sacubitril/valsartan compared to valsartan. Before double-blinding, the participants underwent run-ins of single-blind periods where tolerability to both the drugs was assessed. The functional capacity and quality of life were measured by the Kansas City Cardiomyopathy Questionnaire (KCCQ) and NYHA (New York Heart Association) classifications. There was a change in clinical summary score (CSS) from Baseline to month 8 for the patients but the results were not statistically significant. The superiority test revealed the least squares mean of difference was 1.02, 95% CI (95% $-0.0047$ to 2.0576, *p*-value 0.0510). The study also focused on a change of NYHA class from baseline to month 8 and found superiority for the drug with statistical significance with an odds ratio of 1.4457 (95% CI (1.1294 to 1.8552), *p*-value of 0.0035). The drug reduced risk of hospitalization in women more than that in men. The change in NYHA were similar in both genders but KCCQ improvement was less in females. This may indicate sex disparity in outcomes of this therapy. The strict inclusion criteria may present problems with the external validity of these findings. The use of active comparator valsartan was a great idea by the investigators since it is widely used in HFpEF patients. However, the trial has a requirement for an elevated natriuretic peptide which may exclude patients without such elevation. Also, patients with BMI > 40 kg/m$^2$ were excluded, which again will limit generalizability in obese populations.

### 3.2. NCT03066804 [13]

The PARALLAX trial was a 24-week RCT compared drug therapy with individual comparators such as enalapril, valsartan, or placebo. The trial was conducted in 396 centers and 32 countries. It focused on the exercise, functional capacity, and quality of life more closely with primary endpoints of change from baseline in plasma NT-proBNP level at week 12 and in the 6-min walk distance at week 24. It categorized KCCQ out-

comes with its deterioration and improvement over time by five points. Furthermore, it also included quality of life measures such as the NYHA classification and I short form 36 health survey (SF-36) physical component summary (PCS) Score at Week 24. A total of 2566 patients were studied, 1281 were prescribed sacubitril-valsartan and another 1285 were provided individualized medical therapy (IMT). There was no change from baseline in the 6 Minute Walk Distance (6MWD) at Week 24 in the sacubitril-valsartan vs. IMT group. Mean difference −2.49 95% CI (−8.5267 to 3.52297, *p*-value 0.416). The mean change from baseline in KCCQ score at Week 24 was not significant. The mean difference in the score in the sacubitril-valsartan vs. IMT group was 0.523 95% CI (−0.9258 to 1.9720, *p*-value 0.479). Percentage of Patients With ≥ 5-points Deterioration in KCCQscore at Week 24 was not significant OR 1.1 95% CI (0.83, 1.47, *p*-value 0.49). The change from baseline in the NYHA functional class at Week 24 was not significant OR 0.979 95% CI (0.81, 1.18, *p*-value 0.83). The change from baseline Ihe SF-36 PCS score at Week 24 was not significant (mean difference −0.15 95% CI (−0.8093 to 0.4953, *p*-value 0.63)). The trial also revealed that there were more adverse events such as hypotension, hyperkalemia, and albuminuria in the sacubitril-valsartan group.

*3.3. NCT00887588 [14]*

The PARAMOUNT was a 12-week double-blind RCT comparing outcomes of change in NT-proBNP and quality of life in sacubitril-valsartan vs. valsartan only groups. The primary outcome of interest was a change in NT-proBNP which revealed a significant reduction in serum levels. Quality of life was assessed by several methods. Change from baseline in KCCQ, overall Summary Score and individual domain summary Scores were calculated for each group. In the study, the KCCQ Scores were analyzed by least squared mean and it showed a reduction in physical limitation symptom stability symptom burden social limitation and overall summary score. However, the least squared mean values were increased in symptom frequency, self-efficacy, total symptom score, and quality of life. The percentage of participants with clinical composite assessment of: improved, unchanged, or worsened. The composite score improved by 41.7% in the LCZ696 group compared to the valsartan group where only 32.8 percent showed improvement. The percentage of participants with NYHA Class I, II, II, or IV showed the stratification of patients in LCZ696 vs. valsartan group. At week 36, both groups showed improvement in symptom profiles. This trial also extensively studied heart strain imaging and revealed impaired systolic function despite preserved global LVEF (Left ventricular ejection fraction) in HFpEF. Until now, diastolic dysfunction was a known culprit in HFpEF but findings of the imaging results indicate that systolic dysfunction could also be a contributor to its pathophysiology.

## 4. Discussion

Sacubitril-valsartan combination is of recent interest in the treatment of heart failure and it has proved to be superior to the likes of ACE (Angiotensin-converting enzyme) inhibitors in some studies [15]. We evaluated the data on the drug and its effect on functional capacity, quality of life, and exercise capacity. The trials used various measures of functional capacity with questionnaires such as KCCQ, tools such as clinical improvement scores, categorical classification such as NYHA class, and tests such as the 6-min walking distance. However, no study used peak oxygen uptake, ventilation/carbon dioxide ratio slope, or other measures that could better quantify the effect on exercise capacity due to sacubitril-valsartan. The European association of preventive cardiology has laid down an appraisal of exercise testing in HFpEF. It also highlights that objective measures of exercise capacity have not been included in any diagnostic algorithms. The measures of gas exchange analysis by cardiopulmonary exercise testing (CPET) are considered gold standard for non-invasive functional capacity evaluation. CPET also can be utilized to identify non-cardiac causes of dyspnea [16]. For instance, COPD is one of the most common comorbidities found in HFpEF patients. In such cases, whether the dyspnea is due to COPD or heart failure symptoms is often confusing to physicians. The trails in our review do not

perform CPET for differentiation. Additionally, no study has attempted to categorize the HFpEF phenotype and there is a lot of room for improvement here since there is increasing interest in phenotype-specific treatments of HFpEF.

Although 6MWD provides prognostic information, only one study examined the said outcome. Furthermore, the 6MWD has limited predictive value and is often insensitive to minor improvements [17]. The studies in our review that evaluated daily physical activity in HFpEF patients who were on sacubitril-valsartan showed no significant improvement [18]. Our review indicates that the functional capacity, exercise capacity, and quality of life of patients have little to no change due to the use of sacubitril-valsartan and more studies focused on the mentioned outcomes should be performed. It is also possible a better classification of a heterogeneous disease such as HFpEF is needed while conducting clinical trials [19]. Improvement in quality of life and exercise capacity are important treatment targets. Some researchers recommend endpoints like days alive and out of the hospital [20] because reductions in such parameters severely impact patient's day-to-day life and burden. Quality-adjusted life years (QALY) is another useful parameter to determine the quality of life [21]. Other tools that may be useful in determining functional capacity include the Minnesota Living with Heart Failure Questionnaire (MLHFQ), the EQ-5D, Chronic Heart Failure Assessment Tool, the Cardiac Health Profile congestive heart failure, the Chronic Heart Failure Questionnaire, the Left Ventricular Disease Questionnaire, and the Quality of Life in Severe Heart Failure Questionnaire [22]. These tools have been infrequently used in HFpEF. It is well documented that exercise training improves functional capacity in patients with HFpEF [23]. With regards to exercise limitations, proper recognition and measurements for cardiac and noncardiac contribution to the limitation must be considered.

Hence, future studies could also focus on the synergistic impact of pharmacotherapy and exercise in HFpEF. Also, for studies to yield relevance to real life, the definition and diagnostic criteria of HFpEF also must be standardized. Currently, several criteria exist which adds to the confusion and difference in inclusion and exclusion criteria of patients. There have been efforts to even use machine learning in its diagnosis [24]. Angiotensin-neprilysin inhibition has been of continued interest due to its role in physiologic control of cardiovascular function [25] and its proven benefit with mortality as well as hospitalizations in clinical trials has led to a need for more evidence in other outcomes of the drug.

Limitations of our study include a lower number of RCTs, all included studies being industry-funded (Novartis), and only one of the studies with a primary outcome related to functional capacity, quality of life, or exercise capacity. Another limitation is that no studies have classified patients keeping in mind heart failure with mid-range ejection fraction (HFmEF) in their studies which is a relatively new classification of heart failure but has been known to have distinct characteristics, response to therapy, and prognosis [26]. More standardized RCTs with larger scale, longer intervention measures, and measures specific to functional/exercise capacity and quality of life must be conducted to address this.

Strengths of our review include a rigorous review of literature for clinical trials studying the quality of life and exercise capacity, the inclusion of clinical trials of the highest quality, robust study design of the trials, [27] availability of data from each study, and large sample sizes.

## 5. Conclusions

Previous large-scale studies have mostly focused on the effect of pharmacotherapy on morbidity and mortality of heart failure with preserved ejection fraction. Since, thechronic functional capacity is an important factor when considering treatment, this systematic review was carried out to evaluate the association of sacubitril-valsartan with the quality of life, exercise and functional capacity of patients with HFpEF.

The review will prove to help cardiologists who are ambivalent about the use of sacubitril-valsartan on the short-term outcomes and functional capacity of the patients. It also highlights the need of using recommended methods used to study these outcomes by the trials. Hopefully, this review will also benefit researchers as a critique to current

trials and a roadmap for conducting robust trials if they seek to perform studies on quality of life, exercise and functional capacity in HFpEF. The current evidence points towards the minimal to no impact of the drug combination on the short-term functional outcome of the disease. However, due to the lower number of RCTs performed the hypothesis should be validated by further high-quality RCTs focused on the primary outcome of the exercise capacity.

**Author Contributions:** Conceptualization, A.V., C.V., H.F., A.A. and A.S.; methodology, A.V., I.N., Z.A. and C.V.; validation, S.K., M.A., S.M. and H.F.; formal analysis, A.V. and A.S.; data curation, I.N., M.A., S.M., Z.A., A.A. and S.K.; writing—original draft preparation, A.V., A.S., C.V., H.F., I.N., M.A., S.M., Z.A., A.A. and S.K.; writing—review and editing A.V. and S.K.; supervision, A.V. and S.K.; project administration, A.V. All authors have read and agreed to the published version of the manuscript.

**Funding:** This research received no external funding.

**Institutional Review Board Statement:** Not applicable.

**Informed Consent Statement:** Not applicable.

**Data Availability Statement:** Data was procured from publicly available data on databases and the trial data from ClinicalTrials.gov (accessed on 20 July 2022).

**Conflicts of Interest:** The authors declare no conflict of interest.

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
