# Peer review of "Effect of Sacubitril-Valsartan on Quality of Life, Functional and Exercise Capacity in Heart Failure with Preserved Ejection Fraction (HFpEF): A Systematic Review of Randomized Clinical Trials"

_hearts, doi:10.3390/hearts3040015_

Round 1

Reviewer 1 Report

Author has reviewed the article in reference to improvement in the functional efficacy of HFpEF patients on the sacubitril-valsartan protocol. However, this review has many pitfalls to cover.

1. Author should give complete information on the selection of databases that narrowed down to n=3 for review.

2. In review article, author should detail the demographics of patients and comorbidities e.g. DM and HTN. This can give more insighful view of HFpEF and impact of ARNI

3. Author should provide distinctive conclusion view from already conducted trial conclusion.

Author Response

Thank you for reviewing my manuscript. I greatly appreciate your time and input in critically appraising out review. 

I have considered your suggestions and made major changes to my document. 

  1. We performed a comprehensive broad search again citing other reviewer's suggestions to include additional databases. Following that, we took your suggestion and included individual numbers from all the databases and also mentioned reasons for exclusion in the diagram as well as the manuscript.
  2. Considering the demographics was a great idea and we identified sex disparity in one of the trials which we have included in our review. Thank you for this suggestion.
  3. We edited the conclusion as per your suggestion to indicate how this review serves critique of current trials and as a roadmap for clinicians as well as researchers to conduct robust clinical trials to accurately measure the quality of life, functional capacity, and exercise capacity. 

Reviewer 2 Report

1. Why was not registered in the PROSPERO? 

2. Why were not EMBASE and SCOPUS  used? 

3. You should correct abbreviations, sometimes a specific abbreviation was already spelled out and you repeat it again. 

4. Table 2 needs to be improved or removed. 

5. ''There’s also some evidence that cardiomyopathy remodeling may be the underlying cause of the condition. Many other mechanisms are defined but most of them I yet to be validated'' - This frase does not make sense to me. 

6. Said outcomes - I would say ''mentioned outcomes''

7.  why was not SGLT2i studied instead? it has shown to be more effective in HFpEF. 

8. You are using abbreviations such as - Till, doesn't, there's -- They should not be used. Spell out 

9. Using capital letters where you are not suppose to, please correct 

Author Response

Thank you for critically appraising our article.  I appreciate the time and consideration in reviewing our article. Your comments were very useful in majorly revising the core parts of the review and the changes we adopted using all your suggestions have been described below.

  1. Although PROSPERO wasn't used as the review had already begun, we maintained an unbiased procedure for the study by following Cochrane library's guidelines and PRISMA protocol for screening the studies.
  2. EMBASE and SCOPUS weren't used due to lack of access but we revised our search strategy and included Google Scholar and also re-performed a broad search to ensure we don't miss anything. We conducted a search that was comprehensive with the search terms and methodology used. We also included numbers from individual databases and reasons for the exclusion as well.
  3. Abbreviations were corrected and repeated abbreviations were removed
  4. Table 2 was revamped to highlight study characteristics
  5. The phrase was modified by giving additional context in the previous sentence. Thank you for pointing this out. I believe it is more clear.
  6. Changed said outcomes to mentioned outcomes.
  7. Sacubitril valsartan was well-studied in HFrEF and we wanted to explore this drug's effect on HFpEF.
  8. abbreviations such as "doesn't", "there's" were removed and spelled out.
  9. The capitalization was corrected by reviewing the entire document. 

Again, thank you for reviewing our article. 

Reviewer 3 Report

1.     The title of the manuscript is too long, so it is suggested to reduce the number of words.

2.     Tables 1 and 2 should be changed to standard three-line tables.

3.     The words on Figure 1 are blurred and need improved resolution.

4.     Many of the statements in the manuscript are non-standard and revision by a professional is recommended.

5.     It is recommended that the full text be aligned with both ends, not left.

6.     In line 36, “the ejection fraction is normal is no longer viable”; two verbs, please modify.

7.     In line 43, “most of them I yet to be validated”; faulty wording, please modify.

8.     In line 86, “In none of the studies”; two spaces have been entered before this sentence, please modify.

9.     In line 188, “HFrEF” please change to “HFpEF”.

Author Response

Thank you for critically appraising our article. I appreciate your time in reviewing our article. Your comments were immensely useful in the revision and the changes we made using all your comments have been described below.

  1. The title was shortened as suggested.
  2. Tables 1 and 2 were modified and borders were highlighted. 
  3. After revising the document in accordance with what other reviewer's suggested, we created the image again and keeping in mind your suggestion, we created a high-resolution image for our review and included numbers from individual databases. 
  4. We reviewed the manuscript and edited the statements that were ambiguous. 
  5. We edited the entire document to be aligned at both ends.
  6. line 36 was edited and the error was fixed. Thank you for pointing this out
  7. The faulty wording in line 43 was modified.
  8. In line 86, two spaces were removed before the sentence.
  9. In line 188, HFrEF was changed to HFpEF 

Again, thank you very much for the review. 

Round 2

Reviewer 1 Report

Author has addressed comments as per previous comments. 

Reviewer 2 Report

Good Job. 

Reviewer 3 Report

The manuscript has been carefully revised and is recommended for acceptance.